# Large Scale Structure of Neural Network Loss Landscapes

**Stanislav Fort**[*]
Google Research
Zurich, Switzerland

**Stanislaw Jastrzebski**[†]
New York University
New York, United States

## Abstract

There are many surprising and perhaps counter-intuitive properties of optimization of deep neural networks. We propose and experimentally verify a unified phenomenological model of the loss landscape that incorporates many of them. High dimensionality plays a key role in our model. Our core idea is to model the loss landscape as a set of high dimensional *wedges* that together form a large-scale, inter-connected structure and towards which optimization is drawn. We first show that hyperparameter choices such as learning rate, network width and $L_2$ regularization, affect the path optimizer takes through the landscape in similar ways, influencing the large scale curvature of the regions the optimizer explores. Finally, we predict and demonstrate new counter-intuitive properties of the loss-landscape. We show an existence of low loss subspaces connecting a set (not only a pair) of solutions, and verify it experimentally. Finally, we analyze recently popular ensembling techniques for deep networks in the light of our model.

## 1 Introduction

The optimization of deep neural networks is still relatively poorly understood. One intriguing property is that despite their massive over-parametrization, their optimization dynamics is surprisingly simple in many respects. For instance, Li et al. [2018a] show that in spite of the typically very high number of trainable parameters, constraining optimization to a small number of randomly chosen directions often suffices to reach a comparable accuracy. Fort and Scherlis [2018] extend this observation and analyze its geometrically implications for the landscape; Goodfellow et al. [2014] show that there is a smooth path connecting initialization and the final minima. Another work shows how it is possible to train only a small percentage of weights, while reaching a good final test performance [Frankle and Carbin, 2019].

Inspired by these and some other investigations we propose a phenomenological model for the loss surface of deep networks. We model the loss surface as a union of $n$-dimensional (lower dimension than the full space, although still very high) manifolds that we call *n-wedges*, see Figure 1. Our model is capable of giving predictions that match previous experiments (such as low-dimensionality of optimization), as well as give new predictions.

First, we show how common regularizers (learning rate, batch size, $L_2$ regularization, dropout, and network width) all influence the optimization trajectory in a similar way. We find that increasing their regularization strength leads, up to some point, to a similar effect: increasing width of the radial tunnel (see Figure 2 and Section 3.3 for discussion) the optimization travels. This presents a next step towards understanding the common role that different hyperparameters play in regularizing training of deep networks [Jastrzębski et al., 2017, Smith et al., 2018].

---

[*]This work was done as a part of the Google AI Residency program.
[†]This work was partially done while the author was an intern at Google Zurich.

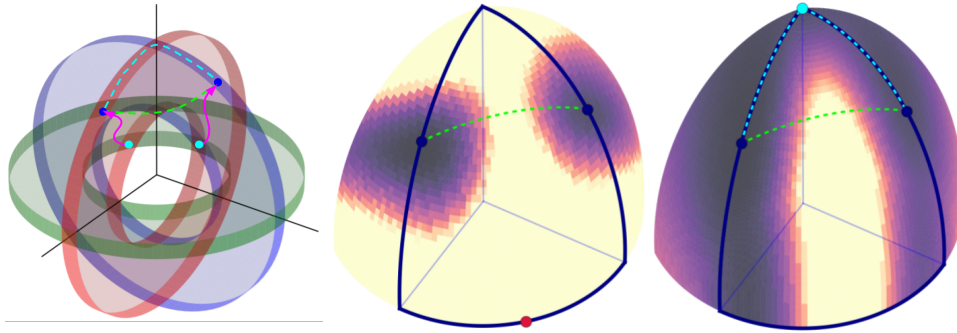

Figure 1: A model of the low-loss manifold comprising of 2-wedges in a 3-dimensional space. Due to the difficulty of representing high dimensional objects, the closest visualization of our landscape model is built with 2-dimensional low-loss wedges (the disks) in a 3-dimensional space (**Left panel**). The particular details such as angles between wedges differ in real loss landscapes. Optimization moves the network primarily radially at first. Two optimization paths are shown. A low-loss connection, illustrated as a dashed line, is provided by the wedges' intersections. In real landscapes, the dimension of the wedges is very high. In the **Central panel**, the test loss on a two dimensional subspace including two independently optimized optima (dark points) is shown (a CNN on CIFAR-10), clearly displaying a high loss wall in between them on the linear path (lime). The **Right panel** shows a subspace including an optimized midpoint (aqua) and the low loss 1-tunnel we constructed connecting the two optima. Notice the large size (compared to the radius) of the low-loss basins between optima.

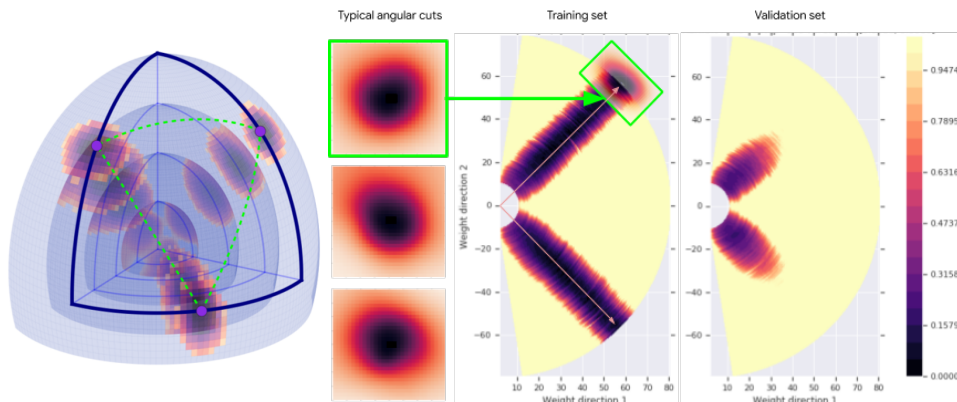

Figure 2: A view of three individually optimized optima and their associated low-loss radial tunnels. A CNN was trained on CIFAR-10 from 3 different initializations. Along optimization, the radius of the network's configuration grows radially. The figure shows experimentally observed training set loss in the vicinity of the three optima at 4 different radii (4 different epochs). The lime lines show that in order to move from one optimum to another on a linear interpolation, we need to leave the low loss basins surrounding optima. Individual low-loss radial tunnels are approximately orthogonal to each other. Note that in high dimensions, the neighborhood for a point in our $n$-wedge (see Figure 1) will *look* like a tunnel locally, while its large scale geometry might be that of a wedge (Section 3.3).

Most importantly, our work analyses new surprising effects high dimensions have on the landscape properties and highlights how our intuition about hills and valleys in 2D often fails us. Based on our model, we predict and demonstrate the existence of higher dimensional generalizations of low loss *tunnels* that we call $m$-tunnels. Whereas a tunnel would connect two optima on a low loss path, $m$-tunnels connect $(m + 1)$ optima on an effectively $m$-dimensional hypersurface. We also critically examine the recently popular Stochastic Weight Averaging (SWA) technique [Izmailov et al., 2018] in the light of our model.

## 2 Related work

A large number of papers show that the loss landscape, despite its large dimension $D$, exhibits surprisingly simple properties. These experiments, however, often do not attempt to construct a single model that would contain all of those properties at once. This is one of the key inspirations for constructing our theoretical model of the loss surface – we want to build a *unified* framework that incorporates all of those experimental results and can make new verifiable predictions. In the following, we present relevant observations from literature and frame them in the light of our model.

**Long and short directions**    First, the linear path from the initialization to the optimized solution typically has a monotonically decreasing loss along it, encountering no significant obstacles along the way. [Goodfellow et al., 2014] To us, this suggests the existence of *long directions* of the solution manifold – directions in which the loss changes slowly. On the other end of the spectrum, Jastrzębski et al. [2018] and Xing et al. [2018] characterize the shape of the sharpest directions in which optimization happens. While Goodfellow et al. [2014] observes a *large scale* property of the landscape, the other experiments are inherently local.

**Distributed and dense manifold**    Another phenomenon is that constraining optimization to a random, low-dimensional hyperplanar cut through the weight space provides comparable performance to full-space optimization, given the dimension of the plane $d$ is larger than an architecture- and problem-specific intrinsic dimension $d_{\text{int}}$. [Li et al., 2018a] The performance depends not only on the hyperplane's dimension, but also on the radius at which it is positioned in the weight space. The results are stable under reinitializations of the plane. [Fort and Scherlis, 2018] This suggests the special role of the radial direction as well as the distributed nature of the low-loss manifold. Put simply, good solution are everywhere and they are distributed densely enough that even a random, low-dimensional hyperplane hits them consistently.

**Connectivity**    Finally, a pair of two independently optimized optima has a high loss wall on a linear weight-space interpolation between them (as well as on any $(P \propto 1 - k/D)$ random path). However, a low-loss connector can be found between them, such that each point along such a path is a low-loss point itself. [Draxler et al., 2018, Garipov et al., 2018] This suggests *connectivity* of different parts of the solution manifold.

**Loss surface of deep networks**    In this paper we present a new model for the loss surface of deep networks. Theoretical work on the subject was pioneered by Choromanska et al. [2015]. An important finding from this and follow-up work is that all minima in the loss surface are in some sense global [Nguyen and Hein, 2017]. Some papers have also looked at ways of visualizing the loss surface [Goodfellow et al., 2014, Keskar et al., 2017, Li et al., 2018b]. However, as we demonstrate, those often do not capture the properties relevant to the SGD, as they choose their projection planes at random.

An important feature of the loss surface is its curvature. One of the first studies on curvature were carried out by LeCun et al. [1998], Sagun et al. [2016]. A significant, though not well understood, phenomenon is that curvature correlates with generalization in many networks. In particular, optimization using a lower learning rate or a larger batch-size tends to steer optimization to a both sharper, and a better generalizing, regions of the loss landscape Keskar et al. [2017], Jastrzębski et al. [2017]. Li et al. [2018b] also suggest that overall smoothness of the loss surface is an important factor for network generalization.

## 3 Building a toy model for the loss landscape

In this section we will gradually build a phenomenological toy model of the landscape in an informal manner. Then we will perform experiments on the toy model and argue that they reproduce some of the intriguing properties of the real loss landscape. In the next section we will use these insights to propose a formal model of the loss landscape of real neural networks.

### 3.1 Loss landscape as high dimensional wedges that intersect

To start with, we postulate that the loss landscape is a union of high dimensional manifolds whose dimension is only slightly lower than the one of the full space $D$. This construction is based on the key surprising properties of real landscapes discussed in Section 2, and we do not attempt to build it up from simple assumptions. Rather, we focus on creating a phenomenological model that can, at the same time, reconcile the many intriguing results about neural network optimization.

To make it more precise, let us imagine a simple scenario – there are two types of points in the loss landscape: *good*, low loss points, and *bad*, high loss points. This is an artificial distinction that we will get rid of soon, but will be helpful for the discussion. Let $n$ of their linear dimensions be *long*, of *infinite* linear extent, and $D - n$ dimensions *short*, of length $\varepsilon$. Let us construct a very simple toy model where we take all possible $n$-tuples of axes in the $D$-dimensional space, and position one cuboid such that its long axes align with each $n$-tuple of axes. We take the union of the cuboids corresponding to all $n$-tuples of axes. In such a way, we tiled the $D$-dimensional space with $n$-long-dimensional objects that all intersect at the origin and radiate from it to infinity. We will start referring to the cuboids as $n$-wedges now[3]. An illustration of such a model is shown in Figure 1. Since these objects have an infinite extent in $n$ out of $D$ dimensions (and $s = D - n$ short directions), they necessarily intersect. If the number of short directions is small $s \ll D$, then the intersection of two such cuboids has $\approx 2s$ short directions.

Even such a simplistic model makes definite predictions. First, every low loss point is connected to every other point. If they do not lie on the same wedge (as is very likely for randomly initialized points), we can always go from one to the intersection of their respective wedges, and continue on the other wedge, as illustrated in Figure 1. A linear path between the optima, however, would take us *out* of the low loss wedges to the area of high loss. As such, this model has an *in-built* connectivity between all low loss points, while making the linear path (or any other random path) necessarily encounter high loss areas.

Secondly, the deviations from the linear path to the low loss path should be approximately aligned until we reach the wedge intersection, and change orientation after that. That is indeed what we observe in real networks, as illustrated in Figure 3. Finally, the number of short directions should be higher in the middle of our low loss path between two low loss points/optima on different wedges. We observe that in real networks as well, as shown in Figure 4.

### 3.2 Building the toy model

We are now ready to fully specify the toy model. In the previous section, we discussed informally dividing points in the loss landscape into good and bad points. Here we would like to build the loss function we call the *surrogate loss*. Let our configuration be $\vec{P} \in \mathbb{R}^D$ as before, and let us initialize it at random component-wise. Let $\mathcal{L}_{\text{toy}}(\vec{P})$ denote the surrogate loss for our toy landscape model, that would reach zero once a point lands on one of the wedges, and would increase with the point's distance to the nearest wedge. These properties are satisfied by a simple Euclidean distance to the nearest point on the nearest wedge, which we use as our surrogate loss in our toy model.

More precisely, the way we calculate our surrogate loss $\mathcal{L}_{\text{toy}}(\vec{P})$ is: 1) sort the components of $\vec{P}$ based on their absolute value, 2) take the $D - n$ smallest values, 3) take a square root of the sum of their squares. This simple procedure yields the $L_2$ distance to the nearest $n$-wedge in the toy model. Importantly, it allows us to optimize in the model landscape without having to explicitly model any of the wedges in memory. Here we align the $n$-wedges with the axes, however, we verified that our conclusions are not dependent on this, nor on their exact mutual orientations or numbers.

### 3.3 Experiments on the toy model

Having a function that maps any configuration $\vec{P}$ to a surrogate loss $\mathcal{L}_{\text{toy}}(\vec{P})$, we can perform, in TensorFlow, simulations of the same experiments that we do on real networks and verify that they exhibit similar properties.

**Optimizing on random low-dimensional hyperplanar cuts.** On our toy landscape model, we replicated the same experiments that were performed in Li et al. [2018a] and Fort and Scherlis [2018]. In the two papers, it was established that constraining optimization to a randomly chosen, $d$-dimensional hyperplane in the weight space yields comparable results to full-space optimization, given $d > d_{\text{intrinsic}}$, which is small ($d_{\text{intrinsic}} \ll D$) and dataset and architecture specific.

The way we replicated this on our toy model was equivalent to the treatment in Fort and Scherlis [2018]. Let the full-space configuration $\vec{P}$ depend on within-hyperplane position $\vec{\theta} \in \mathbb{R}^d$ as $\vec{P} = \vec{\theta}M + \vec{P}_0$, where $M \in \mathbb{R}^{d \times D}$ defines the hyperplane's axes and $\vec{P}_0 \in \mathbb{R}^D$ is its offset from the origin.

We used `TensorFlow` to optimize the within-hyperplane coordinates $\vec{\theta}$ directly, minimizing the toy landscape surrogate loss $\mathcal{L}_{\text{toy}}(\vec{X}(\vec{\theta}))$. We observed very similar behavior to the one in real neural networks: we could successfully minimize the loss, given the $d$ of the hyperplane was $> d_{\text{lim}}$. Since we explicitly constructed the underlying landscape, we were able to related this limiting dimension to the dimensionality of the wedges $n$ as $d_{\text{lim}} = D - n$. As in real networks, the random initialization of the hyperplane had very little effect on the optimization. This makes our toy model consistent with one of the most surprising behaviors of real networks. Expressed simply, optimization on random, low-dimensional hyperplanes works well provided the hyperplane dimensions supply at least the number of short directions the underlying landscape manifold has.

**Building a low-loss tunnel between 2 optima.** While we build our landscape in a way that explicitly allows a path between any two low loss points, we wondered if we could construct them the same way we do in real networks (see for instance Draxler et al. [2018]), as illustrated in Figure 3.

We took two random initializations $I_1$ and $I_2$ and optimized them using our surrogate loss $\mathcal{L}_{\text{toy}}$ until convergence. As expected, randomly chosen points converged to different $n$-wedges, and therefore the linear path between them went through a region of high loss, exactly as real networks do. We then chose the midpoint between the two points, put up a tangent hyperplane there and optimized, as described in Section 4. In this way, we were able to construct a low loss path between the two optima, the same way we did for real networks. We also observed the clustering of deviations from the linear path into two halves, as illustrated in Figure 3.

**Living on a wedge, probing randomly, and seeing a hole.** Finally, another property of optimization of deep networks is that along the optimization trajectory, the loss surface looks locally like a valley or a tunnel [Jastrzębski et al., 2018, Xing et al., 2018]. This is also replicated in our model. A curious consequence of high dimensions is as follows. Imagine being at a point $\vec{P}_0$ a short distance from one of the wedges, leading to a loss $\mathcal{L}_0 = \mathcal{L}_{\text{toy}}(\vec{P}_0)$. Imagine computing the loss along vectors of length $a$ in random directions from $\vec{P}_0$ as $\vec{P}_0 + a\hat{v}$, where $\hat{v}$ is a unit vector in a random direction. The change in loss corresponding to the vector length $a$ will very likely be almost independent of the direction $\hat{v}$ and will be *increasing*. In other words, it will look like you are at a bottom of a circular well of low loss, and everywhere you look (at random), the loss increases in approximately the same rate. Importantly, for this to be true, the point $\vec{P}_0$ does *not* have to be the global minimum, i.e. exactly on one of our wedges. We explored this effect numerically in our toy model, as well as observed it in real networks, as demonstrated in Figure 2 on a real, trained network. Even if the manifold of low loss points were made of extended wedge-like objects (as in our model), *locally* (and importantly when probing in random directions), it would look like a *network of radial tunnels*. We use the size of those tunnels to characterize real networks, and the effect of different regularization techniques, as shown in Figure 5.

## 4 Experiments

Our main goal in this section is to validate our understanding of the loss landscape. In the first section we investigate low-loss tunnels (connectors) between optima to show properties of our model. Interestingly, we see a match between the toy model and the real neural networks here. This investigation leads us naturally to discovering *m-connectors* – low loss subspaces connecting a large number of optima at the same time.

Next, we investigate how optimization traverses the loss landscape and relate it to our model. In the final section we show that our model has concrete consequences for the ensembling techniques of

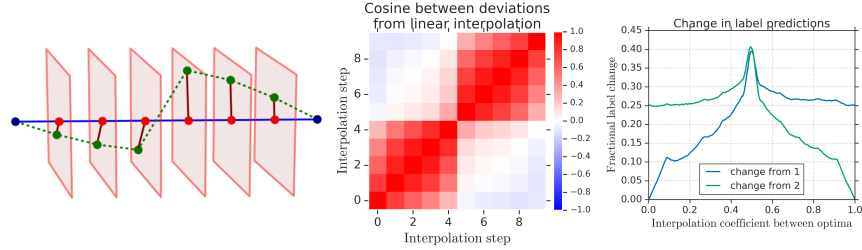

Figure 3: Building a low-loss tunnel. The **Left panel** shows a sketch of our algorithm. Cosines between vector deviations of points on a low-loss connector between optima from a linear interpolation (**Middle panel**). Deviations in the first and second half of the connector are aligned within the group but essentially orthogonal mutually. The **Right panel** shows that label predictions along the tunnel change linearly up to the middle, and stay constant from them on. This is consistent with each wedge representing a particular function. The results were obtained with a CNN on CIFAR-10.

deep networks. All in all we demonstrate that our model of the loss landscape is consistent with both previous observations, as well as is capable of giving new predictions.

## 4.1 Experimental setting

We use SimpleCNN model ($3 \times 3$ convolutional filters, followed by $2 \times 2$ pooling, 16,32,32 channels with the $\tanh$ non-linearity) and run experiments on the CIFAR-10 dataset. Unless otherwise noted (as in Figure 6), we ran training with a constant learning rate with the Adam optimizer.

To further verify the validity of our landscape model, we performed the same experiments on CNN and fully-connected networks of different widths and depths, explored $\mathrm{ReLU}$ as well as $\tanh$ non-linearities, and used MNIST, Fashion MNIST, CIFAR-10 and CIFAR-100 datasets. We do not, however, present their results in the plots directly.

## 4.2 Examining and building tunnels between optima

To validate the model we look more closely at previous observation about the paths between individual optima made in Draxler et al. [2018], Garipov et al. [2018]. Our main idea is to show that a similar connector structure exists in the real loss landscape as the one we discussed in Section 3.3.

In Draxler et al. [2018], Garipov et al. [2018] it is shown that a low loss path exists between pairs of optima. To construct them, the authors use relatively complex algorithms. We achieved the same using a simple algorithm which, in addition, allowed us to diagnose the geometry of the underlying loss landscape.

To find a low-loss connector between two optima, we use a simple and efficient algorithm. 1) Construct a linear interpolation between the optima and divide it into segments. 2) At each segment, put up a $(D-1)$-dimensional hyperplane normal to the linear interpolation. 3) For each hyperplane, start at the intersection with the linear interpolation, and minimize the training loss, constraining the gradients to lie within. 4) Connect the optimized points by linear interpolations. This simple approach is sufficient for finding low-loss connectors between pairs of optima.

This corroborates our phenomenological model of the landscape. In our model we need to switch from one $n$-wedge to another on finding a low-loss connector, and we observe the same alignment when optimizing on our toy landscape. The predicted labels change in the same manner, suggesting that each $n$-wedge corresponds to a particular family of functions. Note also that the number of short directions in Figure 4 at the endpoints (300; corresponding to the original optima themselves) is very similar to the Li et al. [2018a] intrinsic dimension for a CNN on the dataset, further supporting our model that predicts this correspondence (Section 3.3).

## 4.3 Finding low loss m-connectors

In addition, we discover existence of $m$-connectors between $(m+1)$-tuples of optima, to which our algorithm naturally extends. In a convex hull defined by the optima we choose points where we put up

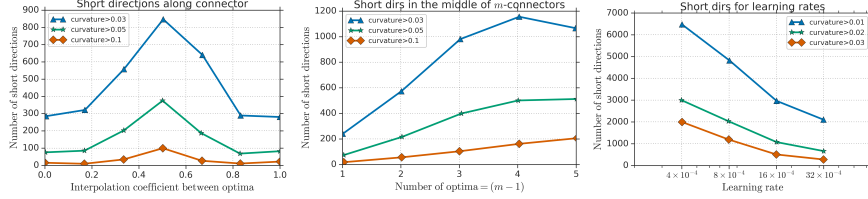

Figure 4: The number of short directions along a 1-connector (tunnel) between 2 optima (Left panel), the number of short directions in the middle of an $m$-connector between $(m + 1)$ optima (Middle panel) and the effect of learning rate on the number of short directions (**Right panel**). The results were obtained with a CNN on Fashion MNIST.

$(D - m)$ dimensional hyperplanes and optimize as before. We experimentally verified the existence of $m$-connectors up to $m = 10$ optima at once, going beyond previous works that only dealt with what we call 1-connectors, i.e. tunnels between pairs of optima. This is a natural generalization of the concept of a tunnel between optima in high dimensions.Another new predictions of our landscape model is that the number of short directions in the middle of an $m$-connector should scale with $m$, which is what we observe, as visible in Figure 4. Note that the same property is exhibited by our toy model. We hope that $m$-connectors might be useful for developing new ensembling techniques in the future.

## 4.4 The effect of learning rate, batch size and regularization

Our model states that optimization is guided through a set of low-loss wedges that appear as radial tunnels on low dimensional projections (see Figure 2). A natural question is which tunnels are selected. While these observations do not directly confirm our model, they make it more actionable for the community.

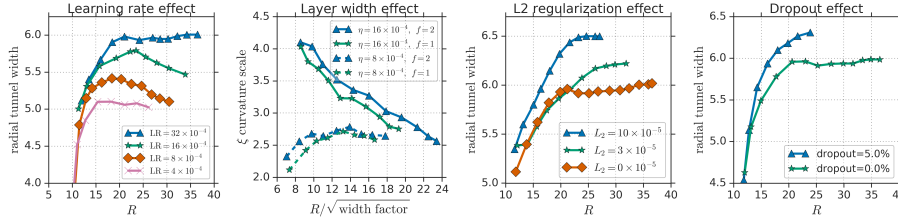

Figure 5: The effect of learning rate, $L_2$ regularization and dropout rate on the angular width of radial low-loss tunnels. The results were obtained with a CNN on CIFAR-10.

We observe that the learning rate, batch size, $L_2$ regularization and dropout rate have a measurable, and similar, effect on the geometrical properties of the radial tunnel that SGD selects. In particular, we observe that the angular width of the low-loss tunnels (the distance to which one can move from an optimum until a high loss is hit) changes as follows: 1) Higher learning rate $\rightarrow$ wider tunnels. As shown in Figure 5. We found the effect of batch size to be of the similar kind, where a larger batch size leads to narrower tunnel. 2) Higher $L_2$ regularization $\rightarrow$ wider tunnels as shown in Figure 5. This effect disappears when regularization becomes too strong. 3) Higher dropout rate $\rightarrow$ wider tunnels as shown in Figure 5. This effect disappears when the dropout rate is too high. We hope these results will lead to a better understanding of the somewhat exchangeable effect of hyperparameters on generalization performance [Jastrzębski et al., 2017, Smith et al., 2018] and will put them into a more geometrical light.

## 4.5 Consequences for ensembling procedures

The wedge structure of the loss landscape has concrete consequences for different ensembling procedures. Since functions whose configuration vectors lie on different wedges typically have the most different class label predictions, in general an ensembling procedure averaging predicted probabilities benefits from using optima from different wedges. This can easily be achieved by

starting from independent random initializations, however, such an approach is costly in terms of how many epochs one has to train for.

Alternative approaches, such as snapshot ensembling Huang et al. [2017], have been proposed. For them, a cyclical learning schedule is used, where during the large learning rate phase the network configuration moves significantly through the landscape, and during the low learning rate phase finds the local minimum. Then the configuration (a snapshot) is stored. The predictions of many snapshots are used during inference. While this provides higher accuracy, the inference is slow as the predictions from many models have to be calculated. Stochastic Weight Averaging (SWA) Izmailov et al. [2018] has been proposed to remedy this by storing the *average configuration vector* over the snapshots.

We predicted that if the our model of the loss landscape is correct, SWA will not function well for high learning rates that would make the configuration change wedges between snapshots. We verified this with a CNN on CIFAR-10 as demonstrated in Figure 6.

The bottomline for ensembling techniques is that practitioners should tune the learning rate carefully in these approaches. Our result also suggests that cyclic learning rates can indeed find optima on different $n$-wedges that provide greater diversity for ensembling, if the maximum learning rate is high enough. However, these are unsuitable for weight averaging, as their mean weights fall outside of the low loss areas.

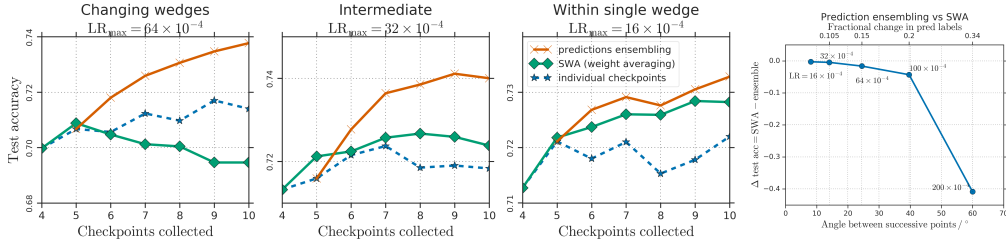

Figure 6: Stochastic Weight Averaging (SWA) does not work well when wedges are changed between snapshots. For high maximum learning rates during cyclical snapshotting, the configurations obtained do not lie on the same wedge and therefore their weight averages lie at a high loss area. The average configuration performs worse than a single snapshots. (**Panel 1**). For a low learning rate, the snapshots lie within the same wedge and therefore their average performs well (**Panel 3**). The advantage predictions averaging (in orange) has over weight averaging (in green) is quantified in **Panel 4**.

## 5 Conclusion

We propose a phenomenological model of the loss landscape of neural networks that exhibits optimization behavior previously observed in literature. High dimensionality of the loss surface plays a key role in our model. Further, we studied how optimization travels through the loss landscape guided by this manifold. Finally, our models gave new predictions about ensembling of neural networks.

We conducted experiments characterizing real neural network loss landscapes and verified that the equivalent experiments performed on our toy model produce corresponding results. We generalized the notion of low-loss connectors between pairs of optima to an $m$-dimensional connector between a set of optima, explored it experimentally, and used it to constrain our landscape model. We observed that learning rate (and other regularizers) lead to optimization exploring different parts of the landscape and we quantified the results by measuring the radial low-loss tunnel width – the typical distance we can move from an optimum until we hit a high loss area. We also make a direct and quantitative connection between the dimensionality of our landscape model and the intrinsic dimension of optimization in deep neural networks.

Future work could focus on exploring impact of our model for more efficient methods for training neural networks.

## Footnotes

[3]This name reflects that in practice their width along the short directions is variable.

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
