[Reviews · NeurIPS 2019]

Reviewer 1



The authors propose a phenemenological model of the loss landscape of DNNs. In particular they devise the landscape as a set of high dimensional wedges whose dimension is slightly lower than the dimension of the full space. The authors first start by building a toy model where they show that the assumptions hold. Afterwards they run experiments with a simple CNN model to show the behavior of the loss landscape. They also show how the optimizer traverses the loss landscape for common hyperparameter choices. Overall I think the paper is well written and the different observations are sound. I like the view the of the loss landscape presented in this paper. I also like the experiments that show the behaviors of the different hyperparameters as well. One thing that would have excited me more is a discussion regarding how this view of the loss landscape helps with training (I know it is hard to comment on immediately). Minor Comments: * Typo in line 8: a similar ways -> similar ways * Line 191: is it SimpleCNN or a simple CNN model? After Author Response: I thank the authors for their response. I appreciate that you have run more experiments and that the results look promising. But at the moment I am not going to update my scores without seeing the results.

Reviewer 2



This paper studies the large-scale structure of neural network objective function (loss landscape). It uses a new idea to not only confirm some known properties of neural network loss landscape but also introduce some new prperties. The authors use the idea of wedges to show three previously known properties of loss landscapes as they call it (1) long and short direction (2) distributed and dense manifold, and (3) connectivity. I should say that because of poor structure of the paper, I could not understand the core part of this paper in which they construct the wedges. All arguments in this paper are based on the concept of wedges. Even though the authors are presenting some nice pictures to make their idea better understood but the text and the pictures are not connected. For example, I do not understand why wedges are disks in Figure 1 when the wedges are supposed to be cuboid (maybe I do not understand the meaning of cuboid here). Another assumption that all other arguments in the paper is built upon is that the loss values stays constant on a wedge. Also the authors are talking about long and short linear dimensions, which I cannot connect. These terms and concepts need to be more explained and defined.

Reviewer 3



This paper takes a unique approach and aims high. In deep learning, there are all these intriguing empirical observations previously known; the road most travelled to understand them is to prove these observations under certain assumptions, while the authors choose to link these observations through a descriptive model that otherwise could have nothing to do with neural networks. I appreciate this unique approach, which is actually a dominant approach in other sciences like physics. The descriptive model in this paper, if more accurate than not, could potentially simplify the conceptual understanding of optimization for deep learning and motivate new algorithms. There are two reasons why I cannot more enthusiastically recommend this paper. 1) As I'm sure the authors understand, the approach taken always runs the risk of overfitting on the previously observed phenomena. To make a quote, all models are wrong but some are interesting. The authors do a good job at showing that the model fits the previously observed phenomena, but this does not convince me that the model interesting. The authors are unable to propose improvements to current optimization algorithms, or make nontrivial and interesting enough new discoveries, at the same level of, say, large loss for linear interpolation vs. low loss along a curve path. 2) The exposition is messy and difficult to understand. I struggle, for example, to understand how the construction in 3.2 gives rise to the structure described in 3.1 and the radial paths in figure 1. I think this is because the authors use mostly words to describe things that can be more precisely described with mathematics. Expressions such as "Cosines between vector deviations of points on a low-loss connector between optima from a linear interpolation" can be very confusing in words. The experimental figures are also confusing. I am still unclear exactly what procedures are used to make those plots. Altogether, I think this work has much potential but is limited by the two problems I outlined. In its current shape, I still think it should be published as a poster just in case it inspires someone.

[Author Response · NeurIPS 2019]

Thank you very much for your detailed reviews and comments. In the rebuttal we will focus on the main issue raised:
lack of clarity in the description of our theoretical model. At the end of the rebuttal we will address the remaining
comments.

**Confusion about our landscape model toy task and the definition of $n$-wedges.** A common point you brought up
is the difficulty in understanding our loss landscape toy task construction, especially what exactly we mean by the
$n$-wedges. We found that in order to be able to verify whether a particular landscape model matched the behaviour
observed in real nets, we needed to implement an explicit simulation.

The simplest version of our toy landscape is constructed as follows. We populate the $D$-dim weight space with $n$-dim
low-loss attractors we call $n$-wedges. Each of these $n$-wedges has $n$ infinitely extended *long* dimensions, and $D - n$
infinitely thin *short* directions. We take each $n$-tuple of axes, and position a single $n$-wedge such that its long directions
are aligned with them. We then define a surrogate loss $\mathcal{L}_{\text{toy}}(\vec{P} \in \mathbb{R}^D)$ for a network configuration $P$ in this weight
space, which we choose to depend monotonically on the $L_2$ distance to the nearest $n$-wedge. Luckily for us, this
distance is simply $d(P) = \sqrt{\text{sum}\left(\text{sorted}(P)[: D - n]^2\right)}$ – an easy to understand explicit expression

While this construction is very specific, we find that it is the dimensions $D$ and $n$ that influence our results, rather than
the specific angles between the $n$-sheets or their axis-alignment. As such, our toy model serves us well, albeit it doesn't
capture many other features of the loss landscape. Nonetheless, on this landscape, we are able to perform connectivity
experiments, as well as experiments with optimizing on random hyperplanes, and empirically verify the similarity to
real network experiments.

In real nets, we find a large number of weight-space directions in which we can move very far, while the loss doesn't
change – those would be the $n$ long directions of the wedge; we also find a small number of extremely sensitive
directions in which a small motion incurs a high loss cost – those are the $D - n$ short directions. Together, these
define locally an $n$-dimensional hyperplane of finite thickness in the remaining $D - n$ thin direction, i.e. a *cuboid*.
Experimentally we notice a strong effect of radius $r^2 = \sum_i w_i^2$, the sum of squares of all weights. While locally a
cuboid, we find that individual parts of the manifold of low loss points radiate from the origin at a well-defined range of
angles, like a *wedge*. We find the full low-loss manifold to be a union of those in different directions and orientations.
We will include this extended discussion in the paper. We will also include an Appendix with a detailed description of
the toy landscape + the code that we use to experiment with it + we will publish a demo Jupyter Notebook / Colab.

**R1: More experiments, larger networks, and harder datasets.** To strengthen the case for our landscape model,
we extended the experiments in our paper to include fully-connected as well as convolutional networks of various
sizes (width, depth, non-linearity) including large models such as the **ResNet20v1** (>90% test on CIFAR-10), trained
on MNIST, Fashion MNIST and CIFAR-10 & 100. To go beyond classification, we also looked at CNN-based
autoencoders. In all cases the results supported our landscape model and we will include them in the final version.
This also demonstrates that our landscape model did not overfit to a small CNN on F-MNIST, as it holds for other
architectures and datasets.

**R5: Overfitting the landscape model to a particular task? New predictions and their empirical observation to
the rescue.** We constructed a model for the loss landscape of neural networks based on existing observations in
literature and our own verification of them. While we were very happy that our model incorporates them all (people in
general had trouble reconciling them together), what gave us confidence were new effects that we predicted based on
the model, that we only *later* observed in real networks. Those were 1) the existence of $(N - 1)$-dimensional low-loss
connectors between $N$-tuples of independent optima, and the scaling of the number of short (=high curvature) directions
in their middle with $N$, 2) the changing of the predicted labels in the middle of a low-loss connector between two
optima, 3) stochastic weight ensembling (SWA) not working when checkpoints are too far from each other (belonging
to different wedges). We were aware of none of those at the time of building our model, and only later we predict they
should happen, and verified them in real networks.

**R2: Getting better at visualizing high-dimensional intuitions in 2D.** During the time between the submission and
now, we developed a better set of figures and explanations to convey the high-dimensional intuitions in 2D and 3D. For
example, we have a better version of Figure 1, where we do not make the wedges circular and smooth, as this was a
confusing illustration for some of our readers.

**R5: Radial tunnels = what low-dimensional cuts would show.** We noticed a confusion about the two types of
"tunnels" we discuss: we use the low-loss connectors between two independent optima as an observation to reconcile
with our model + a diagnostic tool. The other type of a tunnel – the radial tunnel – is what we would see on 2D cuts
through the landscape. At any point in training, making a random, 2D visualization of the loss around our current
point, we would (very likely) see a convex depression. As training progresses mainly radially, and at each point there is
convex depression around us, we can visualize this as a radial tunnel going out. We will be clearer with the distinction
in the final version.

[Meta-Review · NeurIPS 2019]

The authors propose a phenemenological model of the loss landscape of DNNs - they devise the landscape as a set of high dimensional wedges whose dimension is slightly lower than the dimension of the full space, and how the optimizer traverses the loss landscape for common hyperparameter choices. Overall speaking, this paper provide interesting insights to deep learning, although it is not very clear how the insights could be used to improve the training process of deep neural networks yet (to both optimization and generalization). One problem with the paper is its presentation. Some of the reviewers have confusions after reading the paper. It would be critical for the authors to improve their writings (the text and figures) significantly in order to make it more accessible to the audience.